# Dynamic Flow-Adaptive Spectrum Leasing with Channel Aggregation in Cognitive Radio Networks

**DOI:** 10.3390/s20133800

**Published:** 2020-07-07

**Authors:** Xiang Xiao, Fanzi Zeng, Zhenzhen Hu, Lei Jiao

**Affiliations:** 1College of Computer Science and Electronic Engineering, Hunan University, Changsha 410082, China; hdxx@hnu.edu.cn (X.X.); hzz88@hnu.edu.cn (Z.H.); 2College of Information and Electronic Engineering, Hunan City University, Yiyang 413000, China; 3Department of Information and Communication Technology, University of Agder, 4630 Agder, Norway; lei.jiao@uia.no

**Keywords:** cognitive radio networks, flow-adaptive spectrum leasing, channel aggregating

## Abstract

Cognitive radio networks (CRNs), which allow secondary users (SUs) to dynamically access a network without affecting the primary users (PUs), have been widely regarded as an effective approach to mitigate the shortage of spectrum resources and the inefficiency of spectrum utilization. However, the SUs suffer from frequent spectrum handoffs and transmission limitations. In this paper, considering the quality of service (QoS) requirements of PUs and SUs, we propose a novel dynamic flow-adaptive spectrum leasing with channel aggregation. Specifically, we design an adaptive leasing algorithm, which adaptively adjusts the portion of leased channels based on the number of ongoing and buffered PU flows. Furthermore, in the leased spectrum band, the SU flows with access priority employ dynamic spectrum access of channel aggregation, which enables one flow to occupy multiple channels for transmission in a dynamically changing environment. For performance evaluation, the continuous time Markov chain (CTMC) is developed to model our proposed strategy and conduct theoretical analyses. Numerical results demonstrate that the proposed strategy effectively improves the spectrum utilization and network capacity, while significantly reducing the forced termination probability and blocking probability of SU flows.

## 1. Introduction

With the rapid development of wireless communication and the explosive growth of mobile service demand, the under-utilization of the limited wireless spectrum resources has become particularly prominent [1,2]. Cognitive radio networks (CRNs) have been regarded as a promising technology to improve spectrum utilization and alleviate the shortage of spectrum resources [3], where secondary users (SUs) adaptively access the spectrum without affecting transmissions of the primary users (PUs) in a dynamically changing environment [4]. CRNs achieve more flexible and effective spectrum resource allocation through dynamic spectrum access [5].

However, the conventional CRNs always assure the access priority of the PUs, whereas the quality of service (QoS) of the SUs is usually overlooked [4]. Specifically, in overlay CRNs [6], SUs suffer from frequent spectrum handover and forced termination due to PUs’ activities [3]. In underlay CRNs [7], SUs are subject to strict transmission restrictions to avoid interference with PUs [8]. Guaranteeing the QoS for the SU has always been a challenging task [9,10].

The exclusive channel allocation is an effective way to solve the performance degradation of SUs [11,12]. Among existing exclusive channel allocation techniques, spectrum leasing (SL) is an effective approach to improve the SUs’ QoS [13,14], where the PUs lease a part of the licensed spectrum resources to the SUs and gain the appropriate remuneration [15]. Unlike the spectrum access in conventional CRNs, SUs have access priority in the leased spectrum band of leased cognitive radio network (L-CRN). Only in the unleased spectrum band (N-CRN) do the PUs still have the priority of channel access. SL is an effective way to alleviate the degradation of SUs’ performance due to forced termination and transmission restrictions [16].

The cooperative SL schemes [11,17,18,19,20] consider that multiple SUs lease a part of a licensed spectrum band from one PU. Meanwhile, the literature [15,16,21,22,23] analyzes the scenario where multiple PUs provide leased spectrum resources to SUs. In [23], the fixed spectrum leasing (FSL) scheme between PU and SU is investigated. The scope of leased channels in FSL does not change, and such presets degenerate the spectrum utilization in a dynamically changing environment [9]. Different dynamic spectrum leasing (DSL) strategies [24,25,26] have been proposed in the literature. A traffic-adaptive spectrum leasing (TASL) scheme is proposed in [9]. The TASL adjusts the time length of each leasing period according to the amount of the real-time generation secondary packets. Furthermore, a three-dimensional CTMC model is established for the proposed TASL. A three-tier spectrum sharing framework was designed in [24], which allows the network operators to dynamically make two choices: one is to freely access the spectrum of uncertain channel quality, and the other is to lease the spectrum of better channel quality. However, those existing works on DSL in CRNs share some common assumptions. First, the number of channels allocated by each SU is fixed in L-CRN, which usually fails to adapt the real-time demand of heterogeneous users [4]. Second, those strategies adopt exclusive access, which allows PUs to access only unleased spectrum and SUs to access only leased band. The design is not applicable when PUs or SUs demand surges. It is still a research challenge to design spectrum leasing strategies adapted to the dynamic changing environment [27].

In this paper, we propose a novel flow-adaptive spectrum leasing with channel aggregation in multichannel CRNs. Specifically, the proposed SL strategy takes into account the real-time demand changes of users in a dynamic environment, and adaptively adjusts the proportion of leased channels based on the number of ongoing and buffered PU flows. In L-CRN, the flexible spectrum access of SU adopts dynamic channel aggregation, which enables flows to occupy multiple adjacent channels or non-adjacent spectrum holes. Moreover, Energy efficiency is a very important metric in CRNs [28,29]. This work does not include energy efficiency, which is planned as future work. In summary, the main contributions of the paper are outlined as follows:We propose a channel adjustment algorithm for dynamic spectrum leasing, which adaptively adjusts the proportion of leased spectrum according to the amounts of ongoing and buffered PU flows.We adopt the dynamic spectrum access strategy with channel aggregation for the SU flows in the leased spectrum band, which enables each SU flow to occupy multiple spectrum holes based on the available spectrum resource and SU requirements.We employ both priority access and opportunistic access in L-CRN and N-CRN, in order to provide more flexible channel allocation when the demand of flow surges.We develop CTMC to model our proposed strategy and conduct theoretical analyses. The performance for the system is appraised by different metrics. Numerical results demonstrate that the proposed DFSL effectively enhances the network capacity, and improves the forced termination probability and blocking probability of SU flows.

The remainder of this paper is organized as follows. The network scenario is introduced in Section 2. In Section 3 the dynamic spectrum leasing strategy in CRNs and price function are described, followed by Section 4 in which the dynamic spectrum access with channel aggregation is presented. For analyzing the system performance, the continuous time Markov chain (CTMC) model is established in Section 5. In Section 6, the numerical results are explained; thereafter we conclude the paper in Section 7.

## 2. Network Scenario

We consider the infrastructure of a centralized cognitive radio network which consists of a central base station, and multiple PUs and SUs, as illustrated in Figure 1. The central base station plays a role as a coordinator, for example, to obtain the flow information, perform spectrum leasing and allocate spectrum resources. As shown in Figure 2, the licensed spectrum in the CRN is divided into $M\in Z^{+}$ channels, where $Z^{+}$ represents the set of positive integers. The PUs are authorized to access these channels, and can lease a part of licensed channels to the SUs. The leased frequency band is called the leased cognitive radio network (L-CRN). Similarly, the remaining unleased channels are denoted as N-CRN. In the proposed strategy, the number of leased channels, which is denoted by *L*, is adaptively adjusted according to the transmission requirements of the PUs, and the value range is $L\leq L_{\max}<M$, where $L_{\max}\in Z^{+}$ represents the maximum number of channels that the central base station can allocate to the L-CRN.

When spectrum resources are scarce, the flows can be buffered in the queues; the maximum capacity of the PU queue and the SU queue are denoted by $Q_{PU}$ and $Q_{SU}$, respectively. It is worth noting that an important characteristic of our priority queues is that each queue only holds homogeneous services. The flows in the queues follow the first come first served (FCFS) rule.

The traffic flows are sent by the users and can be divided into a series of data packets in the medium access control layer [30,31]. We model traffic at the flow level (at the transportation layer, such as TCP/UDP flows) since modeling at this level captures the dynamics related to the arrival and departure of flows [32]. In our analysis, we assume that the flow arrivals of PUs and SUs follow Poisson processes [30] with rates $\lambda_{P}$ and $\lambda_{S}$, respectively. The service times are exponentially distributed with service rates $\mu_{P}$ and $\mu_{S}$ in one channel. The channels are homogeneous for flows, and the guard band between the channels is ignored. Therefore, the service rate of *i* aggregated channels is $i\mu_{P}$.

Compared to the duration between consecutive service events, the time for channel sensing or spectrum handover is very short, which is ignored in the analysis [30]. It is supposed that the spectrum sensing carried out by SUs has sufficient accuracy and the effect of sensing failure at the flow level is ignorable [33]. It is assumed that flows are independent of each other, and the strategy works with spectrum adaptation. Once accepted, the flow level service will not be terminated due to channel variations with the help of advanced physical and the medium access control layer techniques [30]. Given the above assumptions, we regard the performance obtained from the theoretical analyses as ideal compared with the results based on more realistic conditions.

## 3. Dynamic Adaptive Leased Spectrum Allocation and Price Function

We propose a dynamic flow-adaptive leased spectrum strategy (DFSL), which is distinct from the previous spectrum leasing schemes where the number of leased channels is fixed. DFSL not only meets the spectrum requirements of PUs, but also leases as many channels as possible to SUs, in order to improve spectrum utilization and users’ QoS.

### 3.1. Dynamic Adaptive Leased Channel Adjustment

The key of DFSL in multichannel CRNs is the flexible and variable number of leased channels. In the proposed DFSL algorithm, the number of leased channels adaptively changes according to the number of ongoing and buffered PU flows in the CRN. It means that *L* is not adjusted according to a fixed time point or period. On the contrary, the DFSL algorithm is only executed upon the demand of PU flow changes, such as PU flow arrival or departure events.

Algorithm 1 describes the specific process of adjusting the number of leased channels according to requirements of PU flows. In the algorithm, the requirements of PU flows in the CRN are expressed in four levels: high, medium, low and ultra-low. In order to distinguish these levels, the algorithm sets the corresponding high threshold, middle threshold and low threshold, which are denoted by $\theta_{h}$, $\theta_{m}$ and $\theta_{l}$, respectively, and the value ranges of the thresholds are $1>\theta_{h}>\theta_{m}>\theta_{l}>0$. When the proportion, which is the ratio of the number of ongoing and buffered PU flows to the total number of channels, is within $\left[1,\;\;\theta_{h}\right]$, the corresponding proportion is at a higher level. When the proportion is within $\left[\theta_{h},\;\;\theta_{m}\right]$, the proportion is intermediate. If the proportion is within $\left[\theta_{m},\;\;\theta_{l}\right]$, the proportion is at a low level. Otherwise, the proportion is at an ultra-low level.

The higher the proportion level, the more the PU flows are transmitted and buffered in the CRN, so the fewer the channels that can be leased. The maximum number of leased channels is denoted by $L_{\max}$. When the spectrum resource is sufficient, the numbers of leased channels corresponding to high, medium, low and ultra-low levels are $l\cdot L_{\max}$, $m\cdot L_{\max}$, $h\cdot L_{\max}$ and $L_{\max}$, respectively, where l,m,h(0<l<m<h<1) are the three parameters. In Algorithm 1, ε represents the number of the ongoing PU flows in the unleased channel. η is the number of the ongoing PU flows in the leased spectrum band. Denote ϕ as the number of the buffering PU flows in queue. *M* represents the total number of channels in CRN. L is the number of leased channels for SU. As illustrated in Algorithm 1, the number of adaptive leased channels is limited to $L_{\max}$. The calculation of $L_{\max}$ is described in the following model. When channel resources are scarce, in order to guarantee the QoS requirements of the PUs, the number of leased channels depends on the total number of channels and the number of PU flows in the N-CRN.

When the number of PU flows in the system changes, the flow-adaptive leased channel algorithm is performed to determine *L*. Fortunately the algorithmic complexity of DFSL is very low, and the increase of the number of users and channels will not affect the calculation efficiency. However, the increases of arrival rate and service rate of PU flows will shorten the time interval of implementing the algorithm.
**Algorithm 1:** Flow-adaptive leased channel adjustment algorithm.**Input:** ε, η, ϕ, *M*, $L_{\max}$,**Input:** h,m,l:1>h>m>l>0;**Input:** $\theta_{h},\theta_{m},\theta_{l}:1>\theta_{h}>\theta_{m}>\theta_{l}>0$.**Output:** $L_{}$.1:**if** $\frac{\varepsilon +\eta +\phi }{M}\leq \theta_{l}$ **then**  ▹ The proportion of PU flows is at the ultra-low level.2:  $L_{1}=L_{\max}$3:  **if**  $L_{1}+\varepsilon \leq M$ **then**4:    $L=L_{1}$5:  **else**6:    L=M−ε7:**if** $\theta_{l}<\frac{\varepsilon +\eta +\phi }{M}\leq \theta_{m}$ **then**  ▹ The proportion of PU flows is at the low level.8:  $L_{1}=\left\lfloor h\cdot L_{\max}\right\rfloor$9:  **if** $L_{1}+\varepsilon \leq M$ **then**10:    $L=L_{1}$11:  **else**12:    L=M−ε13:**if** $\theta_{m}<\frac{\varepsilon +\eta +\phi }{M}\leq \theta_{h}$ **then**  ▹ The proportion of PU flows is at the medium level.14:  $L_{1}=\left\lfloor m\cdot L_{\max}\right\rfloor$15:  **if** $L_{1}+\varepsilon \leq M$ **then**16:    $L=L_{1}$17:  **else**18:    L=M−ε19:**if**$\theta_{h}<\frac{\varepsilon +\eta +\phi }{M}\leq 1$ **then**  ▹ The proportion of PU flows is at the high level.20:  $L_{1}=\left\lfloor lL_{\max}\right\rfloor$21:  **if** $L_{1}+\varepsilon \leq M$
**then**22:    $L=L_{1}$23:  **else**24:    L=M−ε25:**return** *L*

### 3.2. Price Function

The price function of the leased channels is analyzed in this subsection. To stimulate SL from the PU’s perspective, there must be adequate remuneration from SU for the leased channels. Therefore, two aspects are taken into account in the design of the pricing scheme. On the one hand, the SL price should be ascending as the number of leased channels increases. On the other hand, with the improvement of the SUs’ QoS, the leasing price must be correspondingly increased [34]. Therefore, the price function is expressed as $P=a{\bar{L}}^{b}$, where $\bar{L}$ represents the average number of leased channels during the spectrum lease period. In this study, $\bar{L}$ is calculated as $\bar{L}=\sum_{X\in S}\sum_{i=1}^{V}j_{l}^{i}r\left(X\right)$, and $0\leq \bar{L}\leq L_{max}$.

*a* and *b* denote the SUs’ QoS in the aspects of $P_{B}^{SU}$ and $P_{F}^{SU}$ respectively. *a* and *b* are non-negative values which are expressed as follows, where α is a scaler for normalization.
(1)$$a=\alpha (1-P_{B}^{SU}),$$
(2)$$b=\left\{\begin{array}{cc} 1+P_{F}^{SU} & \bar{L}<1\\ 2-P_{F}^{SU} & \bar{L}\geq 1 \end{array}\right.,$$

We choose $P_{F}^{SU}$ to be the exponential ingredient on account of the fact that $P_{F}^{SU}$ has a greater impact on users’ QoS than $P_{B}^{SU}$ [35]. Thus, we define the price function as follows: (3)$$P=\alpha \left(1-P_{B}^{SU}\right){\left(\sum_{X\in S}\sum_{i=1}^{V}j_{l}^{i}r\left(X\right)\right)}^{b},$$

From (2) and (3), we can conclude that the leasing reward becomes a higher value when the number of leased channels increases or $P_{B}^{SU}$ and $P_{F}^{SU}$ decrease, and vice versa.

## 4. Dynamic Spectrum Access with Channel Aggregation

In [36], dynamic channel aggregation (DCA) strategies with spectrum adaptation in cognitive radio networks are proposed. However, the spectrum leasing is not taken into account in the design of DCA, which makes it difficult to guarantee SUs’ QoS. The proposed DFSL extends [36] in the adaptive SL and dynamic spectrum access. The DFSL employs a dynamic access mode that combines priority access and opportunistic access. In this way, PU flows can preferentially access N-CRN and opportunistically access L-CRN. Correspondingly, SU flows have the priority over the leased channels and can opportunistically access the unlease spectrum. Each PU flow only occupies one channel. The situation of the SU flows is different. The SU flow adopts channel aggregation with spectrum adaptation [37]. On one hand, when the PU appears on the unleased channel occupied by the SU flow, the ongoing SU flow can jump to another idle channel. On the other hand, the number of aggregated channels for an ongoing SU flow can be adaptively changed according to the number of occupied channels [38].

Different spectrum access strategies in CRNs have been proposed in the literature. Among them, ref. [36] conducted the analysis of the static access strategy, dynamic spectrum access without channel aggregation, the dynamic fully adjustable strategy and the dynamic partially adjustable strategy. The numerical results demonstrate that the dynamic fully adjustable strategy (a,W,V) has significant advantages in higher capacity and lower forced termination probability, where *W* and *V* are the lower limit and the upper limit of aggregation channels for SU flows, respectively. It is also pointed out in [36] that reducing the lower limit or increasing the upper limit can be used to increase the system capacity, and the channel splitting technique is too complicated [30]. Therefore, the proposed dynamic channel access strategy in this paper sets the lower limit of channel aggregation to the smallest positive integer. Moreover, expanding on the dynamic fully adjustable (a,W,V), we propose the waiting queue for the would-be-blocked flows, and $Q_{PU}$ and $Q_{SU}$ are the maximum capacities of buffered PU flows and SU flows, respectively. In the next subsections, the process of flow arrival and departure is introduced.

### 4.1. PU Arrivals

First of all, the flow-adaptive leased channel algorithm is performed to determine *L* when a new PU flow arrives. The channel aggregation and spectrum adaptation are not enabled for PUs. In other words, each PU flow always occupies one channel in the system. Because of the spectrum adaptation for SU traffic mentioned in Section 2, if the number of idle channels in the N-CRN is larger than or equal to one when a new PU flow arrives, the flow starts to transmit and has no impact on the adaptive SUs’ traffic. If there is no idle channel in the N-CRN, the central base station can only forcibly interrupt one ongoing SU flow in the N-CRN for the newly arrived PU flow. If all the unleased channels are already occupied by the PU flows and there are idle leased channels in the L-CRN, the new PU flow traffic can opportunistically access the leased channel. Otherwise, the newly arrived PU flow is buffered in the queue for PU flows. If the queue for PU flows is full, the newly arrived PU flow only can be blocked. The procedure when a new PU flow arrives is illustrated in Figure 3. For simplicity and clarity of expression, in the figures, PU flow and SU flow are denoted by PU and SU, respectively.

### 4.2. SU Arrivals

Differently from PU flow, the SU flow can aggregate multiple channels or occupy one channel for transmission according to channel availability. When a new SU flow arrivals and there is at least one idle channel in the L-CRN, the new SU flow starts transmission. If there is no idle channel in the L-CRN, but there are SU flows in the L-CRN with the number of aggregated channels exceeding 1, the ongoing SU flow with the largest number of aggregated channels donates a channel for the new SU flow. If the number of occupied channels for all SU flows in the L-CRN is 1, and there is at least one PU flow in L-CRN, one of PU flows in the leased spectrum band has to be terminated for the new SU flow. If there are idle channels after all the SU flows have access to the network, the ongoing SU flows in L-CRN can aggregate multiple channels until they reach the upper limit value *V*. If the number of occupied channels for all SU flows in the L-CRN is 1, and there is no idle channel, the new SU flow can access the N-CRN if it is available. Note that the number of unleased channels occupied by the SU flow is fixed as 1, because the SU flow in the N-CRN is at risk of being interrupted by the new PU flow. When there is no idle and available channel, the new SU flow can only be buffered in the queue for SU flows. In the worst case, if there are not enough buffer rooms for SU flow, the newly arrived SU flow only can be blocked. The procedure when a SU flow arrives is illustrated in Figure 4.

### 4.3. PU Departs

First of all, when a PU flow completes the transmission and departs the network, the flow-adaptive leased channel algorithms are used to determine *L*. The departure of PU flow in the N-CRN vacates an unleased channel. If there is an ongoing PU flow occupying the leased channel, the central base station hands over one PU flow from the leased channel to the unleased channel. If there is no PU flow in the L-CRN, a waiting PU flow in the queue accesses the unleased channel. Otherwise, the vacated unleased channel is allocated to the buffered SU in the queue. If there is no buffered flow, the unleased channel only can be temporarily idle.

If a PU flow departs from L-CRN and vacates an unleased channel, it means that there is no buffered SU flow in the queue. The vacated leased channel is allocated to the PU flow buffered in the queue. Otherwise, the vacated leased channel is idle. The procedure when a PU flow departs is illustrated in Figure 5.

### 4.4. SU Departs

When an SU flow in the L-CRN departs, a leased channel is vacated. If there is a SU flow in the queue, the buffered SU flow accesses the vacant leased channel. On the other hand, an ongoing SU flow in the N-CRN is handed over from the unleased channel to the L-CRN. If none of these conditions exist, the vacated channel is allocated to a buffered PU flow in the queue. Otherwise the leased channel is temporarily idle.

When a SU flow in N-CRN completes the transmission and departs from the network, meaning that there is no buffered PU flow at the time, then a buffered SU flow accesses the N-CRN. Otherwise, the vacated unleased channel is temporarily idle. The procedure when a SU flow departs is illustrated in Figure 6.

## 5. CTMC Model and Analysis

### 5.1. CTMC Model for DFSL

In this section, we establish CTMC to model the dynamic flow-adaptive spectrum leasing strategy with channel aggregation in multichannel CRNs. Let *X* represent the general state, and $\left(X=y_{n},j_{n},y_{l},j_{l}^{1},\ldots ,j_{l}^{i},\ldots ,j_{l}^{V},y_{q},j_{q}\right)$, where $y_{n}$ and $y_{l}$ represent the numbers of ongoing PU flow in the unleased spectrum band and the leased channels, respectively. Similarly, $j_{n}^{}$ is the number of SU flows accessed in the unleased spectrum band, and $j_{l}^{i}$ denotes the number of ongoing SU flows with *i* aggregated channels in the L-CRN. $y_{q}$ and $j_{q}$ represent the number of buffered PU flows and SU flows in the queue, and the capacity of the queues for PU flows and the queue for SU flows are $Q_{PU}$ and $Q_{SU}$, respectively. For any state X, the total number of occupied channels in the unleased spectrum band and the leased channels are expressed as $b_{n}\left(X\right)$ and $b_{l}\left(X\right)$, respectively. According to the CTMC, we have $b_{n}\left(X\right)=y_{n}+j_{n}$ and $b_{l}\left(X\right)=y_{l}+\sum_{i=1}^{V}j_{l}^{i}$. The feasible state set of the system is represented as *S*, and $S=\{X|y_{n},j_{n},y_{l},j_{l}^{1},\ldots ,j_{l}^{i},\ldots ,j_{l}^{V},y_{q},j_{q}\geq 0;y_{n}+j_{n}<M-L;y_{l}+\sum_{i=1}^{V}j_{l}^{i}\leq L;y_{q}\leq Q_{PU};j_{q}\leq Q_{SU}$.

As mentioned Section 2, the flow arrivals of PUs and SUs are assumed to follow Poisson processes with rates $\lambda_{P}$ and $\lambda_{S}$, respectively. The service times are exponentially distributed with service rates $\mu_{P}$ and $\mu_{S}$ in one channel, and the service rate of *i* aggregated channels is $i\mu_{P}$. In Algorithm 1, for assuring the QoS requirements of the PU flows in the CRN, the number of dynamically leased channel *L* is limited to $L_{\max}$. The maximum number of unleased channels should not be less than the average expected required by the PU flow, which is expressed as $M-L_{\max}\leq \left\lceil \lambda_{P}/\mu_{P}\right\rceil$. Therefore, $L_{\max}=\left\lceil M-\lambda_{P}/\mu_{P}\right\rceil$ in the proposed DFSL strategy.

The transition rates in a CTMC are defined to be conditioned only on the current state of the process, and the exponential distribution has a memoryless property. Therefore, the procedures when the PU flows arrive/depart in the system are of birth-death process (BDP). In the BDP, the states represent the number of ongoing PU flows in the system. The state transitions have two types; namely, “births” and “deaths”. When birth happens, the number increases by one. On the contrary, when “death” happens, the number decreases by one [38]. The transition rates are indicated in the Figure 7. The procedures when the SU flows arrive/depart can be derived in a similar way. Details of the transitions of DFSL upon a PU flow arrival, a PU flow departure, a SU flow arrival and a SU flow departure can be found in Table 1, Table 2, Table 3 and Table 4, respectively.

### 5.2. CTMC Performance Analysis

In the CTMC performance analysis for DFSL, the stability probability of the state *X* is represented by r(X), where X∈S. $p_{\alpha \beta }$ denotes the state transition rate from one feasible state α to another state β, and $\;0\leq p_{\alpha \beta }\leq 1$. According to the state transitions presented in Table 1, Table 2, Table 3 and Table 4, transition rate matrix *Q* can be established, with each element representing the corresponding transition rate. By the normalization equation and global balance equation of transition rate matrix *Q*, we can calculate the steady state probability r(X) of the Markov model as follows
(4)$$\left\{\begin{array}{l} rQ=0\\ \sum_{X\in S}r(X)=1 \end{array}\right.$$

Once the r(X) are decided from Equation (Equation 4), the performance for the system could be appraised by different metrics. Then, we present the derivations of mathematical expressions in terms of multiple performance metrics.

#### 5.2.1. Spectrum Utilization

In our model and analysis, the ratio, which is the average number of occupied channels for the feasible state over the total number of channels in the CRN, reflects the spectrum utilization of the system [32]. Let $U_{CRN}$ be the spectrum utilization; then we obtain that
(5)$$\begin{array}{ll} U_{CRN} & =\sum_{X\in S}\frac{b(X)}{M}r\left(X\right)\\ \; & =\sum_{X\in S}\frac{b_{n}\left(X\right)+b_{l}\left(X\right)}{M}r\left(X\right)\\ \; & =\sum_{X\in S}\frac{y_{n}+j_{n}+y_{l}+\sum_{i=1}^{V}j_{l}^{i}}{M}r\left(X\right). \end{array}$$

#### 5.2.2. Network Capacity

The network capacity represents the average number of finished flows per time unit. It is expressed by summing up the probability of the stationary of the states and the flow service rate. Denote $\rho_{P}$ and $\rho_{S}$ as the capacity of primary network and the capacity of secondary network, respectively. We have that,
(6)$$\rho_{P}=\sum_{X\in S}\left(y_{n}+y_{l}\right)\mu_{P}r\left(X\right),$$
(7)$$\rho_{S}=\sum_{X\in S}\left(j_{n}+\sum_{i=1}^{V}j_{l}^{i}\right)\mu_{S}r\left(X\right).$$

#### 5.2.3. Blocking Probability

The blocking probability represents the probability that the new flow can only be blocked or dropped when there are not enough available channels in the system. The PU flows can preferentially access N-CRN and opportunistically access L-CRN. So, the PU flow will be blocked only when the following conditions occur at the same time. There is no idle channel in CRN; no ongoing SU flow exists in N-CRN; and the queue for PU flows is saturated. Consequently, the blocking probability of PU flow, $P_{B}^{PU}$, can be expressed as
(8)PBPU=∑X∈S;b(X)=M;jn=0;yq=QPUr(X).

SUs, which lease a part of licensed channels, can preferentially access to the leased channels and opportunistically access the unleased spectrum band. Therefore, the new SU flow will be blocked only if (a) there is no available channel and no aggregation SU flow in CRN, (b) no ongoing PU flow exists in L-CRN and (c) there is no waiting room in the queue for SU flows. The blocking probability of SU flow, $P_{B}^{SU}$, is obtained as
(9)PBSU=∑X∈S;jq=QSU;yl=0;b(X)=M;∀i>1,jli=0.r(X).

#### 5.2.4. Forced Termination Probability

The forced termination probability refers to the probability that the ongoing flow is forced to be interrupted because of the arrival of the other flows with high access priority. In our proposed DFSL strategy, the PU flow, which opportunistically accesses L-CRN, may be interrupted by the new SU flow, when there is no available channel. Let $P_{F}^{PU}$ denote the forced termination probability of PU flow, which is calculated as follows,
(10)PFPU=∑X∈S;yn>0;M=b(X).ylLλS(1−PBPU)λPr(X).

Differently from other spectrum access schemes in CRNs, only the ongoing SU flows in the N-CRN have the risk of being forcibly interrupted. The forced termination probability of the SU flows is represented by $P_{F}^{SU}$; then we obtain that
(11)PFSU=∑X∈S;jn>0;b(X)=M.jnM−Lλp(1−PBSU)λSr(X).

## 6. Performance Evaluations

In this section, the simulation and numerical results are presented to evaluate the performance of the DFSL with channel aggregating. According to [32], we assume a CRN with six channels in the licensed spectrum; i.e., M=6. The arrival rates and service rates for flows are set as $\lambda_{P}=2$, $\lambda_{S}=4$, $\mu_{P}=1$ and $\mu_{S}=1$. The capacities of the queue for PU flows and SU flows are $Q_{SU}=2$ and $Q_{PU}=2$, respectively. The default values for adaptive leased channel adjustment are configured as l=0.3, m=0.6, h=0.8, $\theta_{l}=0.25$, $\theta_{l}=0.5$ and $\theta_{m}=0.75$.

As we model the system performance at flow level, following the common practice of flow-level analysis with CTMC, we address the stochastic behavior of traffic flows using Poisson process for arrivals and exponential distribution for flow durations. Therefore, the simulation is only carried out based on those distributions for validating the correctness of the mathematics model. Indeed, the stochastic nature of mobile radio channel and the uncertainty of interference will influence the statistics of traffic flows. The simulation study of real-measurement-based flow statistics shows that the result of CTMC models can still offer a relative precise reference for the system performance [38]. A more comprehensive system-level simulation is planned as future work. In the following subsections, we investigate the multiple metrics of DFSL through MATLAB calculations and simulations.

### 6.1. Network Capacity

Figure 8 provides the SU network capacity performance comparison between the proposed scheme and the other methods. With higher $\lambda_{P}$, the network capacity of SU flow for the proposed DFSL and the dynamic channel aggregation without spectrum leasing (DCA) have continuously decreased. However, the decrease in network capacity of the proposed DFSL is much smaller than that of the DCA. This is because in DCA, the PU flow has access priority to all channels, and the increased $\lambda_{P}$ may make SU flow unable to access the network or even be forcibly interrupted. In DFSL, the SU flows obtain the priority to access the leased channels. The increased $\lambda_{P}$ can only affect the SU who opportunistically accesses the N-CRN.

From Figure 8, we observe that the SU network capacities in traffic-adaptive spectrum leasing (TASL) and fixed spectrum leasing scheme (FSL) hardly change with the increase of $\lambda_{P}$. This is because the SU network capacities in TASL and FSL are only affected by the number of SU packets and leased channels, respectively. When $\lambda_{P}<4$, the SU network capacity of DFSL is higher than that of TASL. However, TASL obtains better capacity than DFSL does after $\lambda_{P}=4$.

The PU network capacity ($\rho_{P}$) as a function of PU flow arrival rate ($\lambda_{P}$) is plotted in Figure 9. With higher $\lambda_{P}$, the PU network capacity of three strategies initially increases almost linearly. However, the increase rates of $\rho_{P}$ in DFSL and FSL are smaller than that in DCA after $\lambda_{P}=4$. Especially due to the restriction of fixed leased channels, the increase rate of PU network capacity in FSL flattens out. From Figure 8 and Figure 9, we can observe that compared with the channel access strategy without spectrum leasing, both FSL and DFSL improve the capacity of SU flows for the same configuration. However, the capacity of PU has decreased, which is the main side effect of the channel access strategy with spectrum leasing. From Figure 9, we can observe that the loss of PU capacity in FSL is more serious than that of DFSL, when PU flow arrival rate increases. The reason is that although the principle of FSL is simple, the channel allocation cannot be changed according to the requirements of PU.

### 6.2. Blocking Probability

Figure 10 depicts the SU flow blocking probability ($P_{B}^{SU}$) as the PU flow arrival rate ($\lambda_{P}$) varies. With the increase of $\lambda_{P}$, channel resources are gradually scarce. In the proposed DFSL, the blocking probability of the threshold $\theta_{l}=0.25$, $\theta_{m}=0.5$ and $\theta_{h}=0.75$ is higher than the case where the threshold is $\theta_{l}=0.35$, $\theta_{m}=0.7$ and $\theta_{h}=0.85$. For instance, when the PU flow arrival rate is 4 in Figure 10, the SU flow blocking probability is higher by 15% in the system configured as $\theta_{l}=0.35$, $\theta_{m}=0.7$ and $\theta_{h}=0.85$ than that configured as $\theta_{l}=0.25$, $\theta_{m}=0.5$ and $\theta_{h}=0.75$. The reason is that in the flow-adaptive leased channel adjustment algorithm, the smaller the threshold value, the fewer the channels that can be leased when the PU arrival rate increases.

The blocking probability of PU flow ($P_{B}^{PU}$) is illustrated in Figure 11 as $\lambda_{P}$ varies. Obviously $P_{B}^{PU}$ becomes higher as PU flows become more active in three schemes. From Figure 11, we observe that the overall $P_{B}^{PU}$ in DFSL is improved compared to the other schemes. This is because the resource allocation does not adapt to the dynamically changing of PUs demands in FSL and TASL. When $\lambda_{P}>4$, the gap between the three lease schemes is greater, and the disadvantage of $P_{B}^{PU}$ in FSL and TASL is more obvious. For instance, when the PU flow arrival rate is 5 in Figure 11, the PU flow blocking probability is higher by 63.5% in TASL than that in our proposed DSFL.

### 6.3. Forced Termination Probability

The forced termination probability of SU flows ($P_{F}^{SU}$) as a function of the PU flow arrival rate ($\lambda_{P}$) is plotted in Figure 12. With higher $\lambda_{P}$, $P_{F}^{SU}$ in DFSL and DCA has deteriorated. However, the forced termination probability of SU flows in DFSL is much lower than that in DCA. In particular, the increased rate of SU flow forced termination probability flattens out after $\lambda_{P}=3$. The reason is that the PU flow only interrupts the ongoing SU flow in N-CRN, and the SU flows in the leased channels are not affected by PU activities. DFSL has effectively improved the forced termination probability of SU flows.

## 7. Conclusions

In this paper, we propose a novel dynamic flow-adaptive spectrum leasing with channel aggregation (DFSL) for multi-channel CRNs. DFSL provides insights on the design of effective spectrum resource allocation in dynamically changing environments. A leasing algorithm is developed to adaptively adjust the portion of leased channels. In the leased spectrum band, the SU with spectrum priority access adopts the dynamic channel aggregating. The developed analytical models were validated by simulations. The calculations and results demonstrate that the DFSL effectively improves the performance of SU.

## Figures and Tables

**Figure 1 sensors-20-03800-f001:**
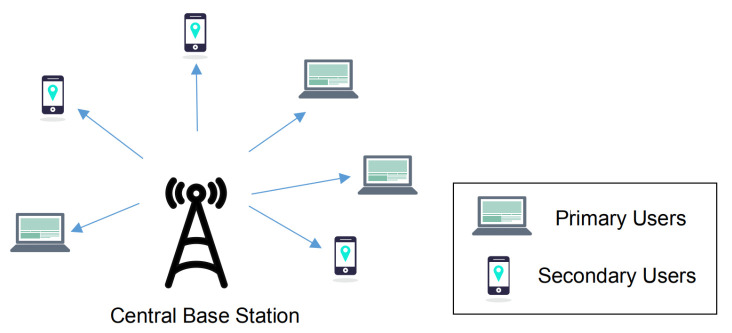
Illustration of the centralized CRN scenario.

**Figure 2 sensors-20-03800-f002:**
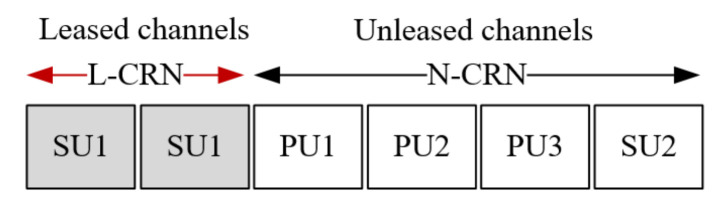
Channel assignment in the cognitive radio network (CRN).

**Figure 3 sensors-20-03800-f003:**
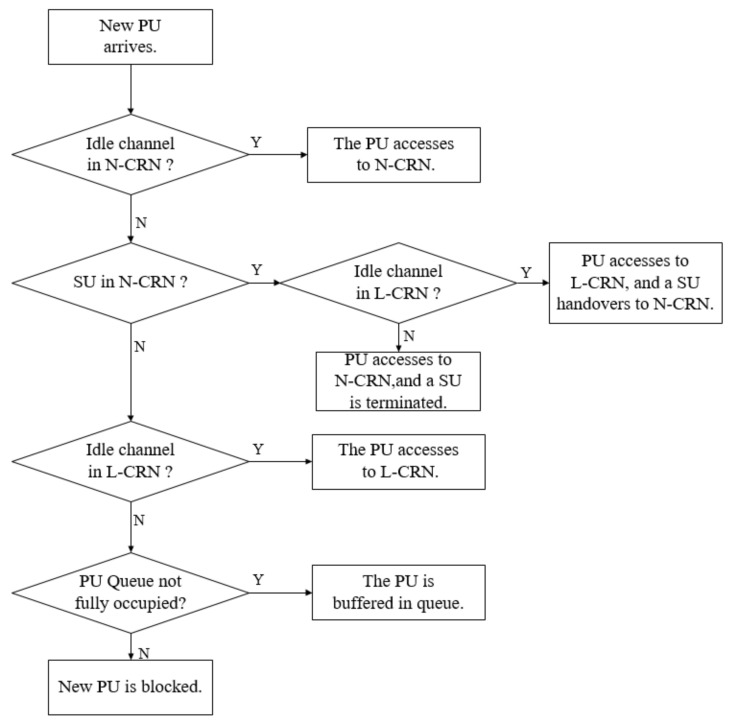
The procedure when a new primary user (PU) arrives.

**Figure 4 sensors-20-03800-f004:**
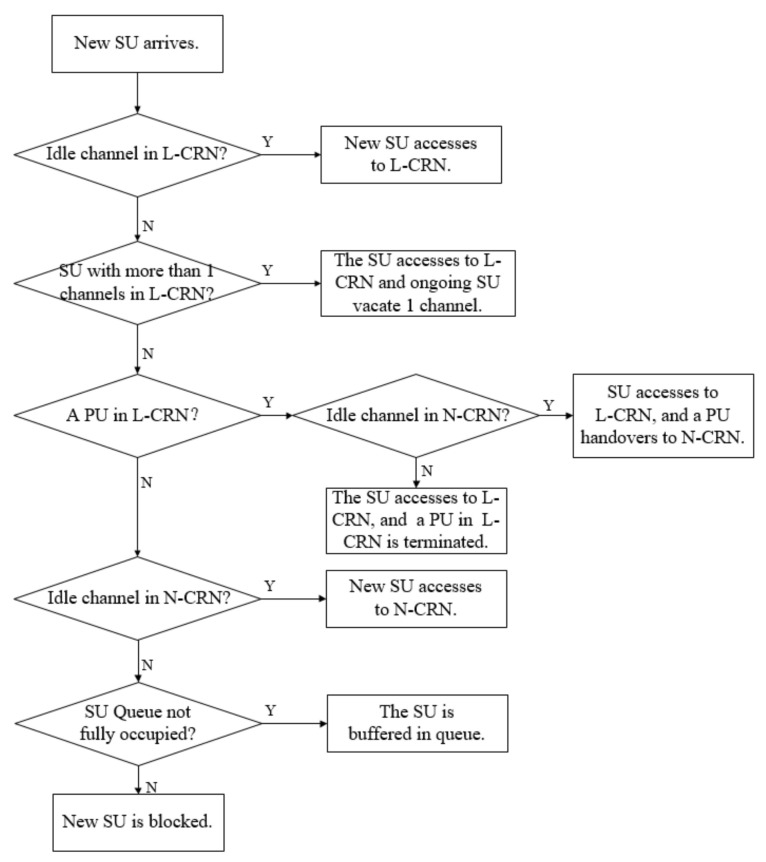
The procedure when a new secondary user (SU) arrives.

**Figure 5 sensors-20-03800-f005:**
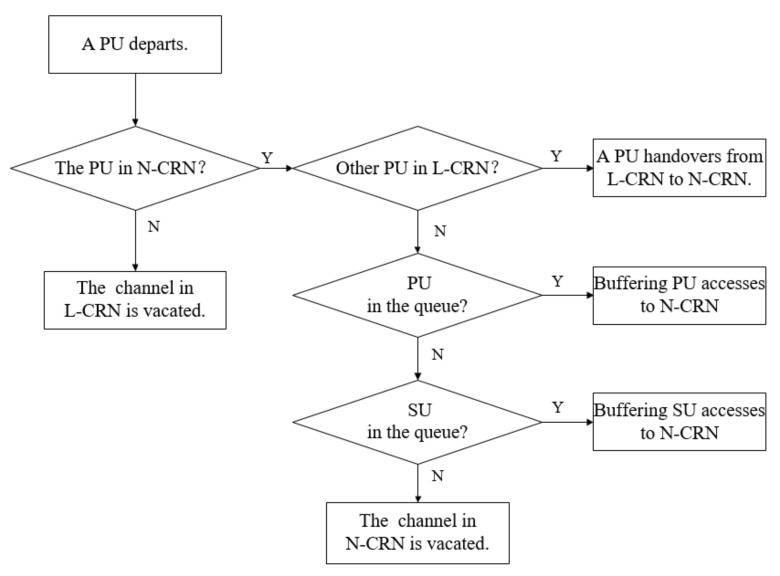
The procedure when a PU departs.

**Figure 6 sensors-20-03800-f006:**
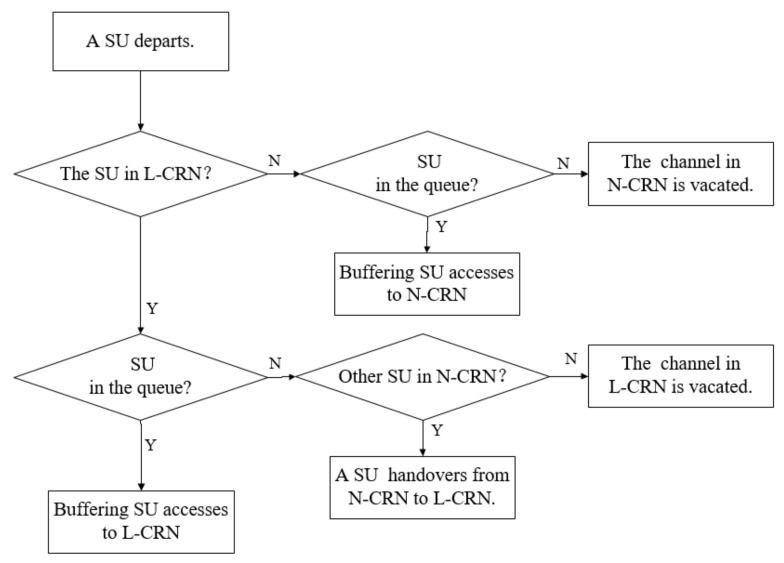
The procedure when a SU departs.

**Figure 7 sensors-20-03800-f007:**
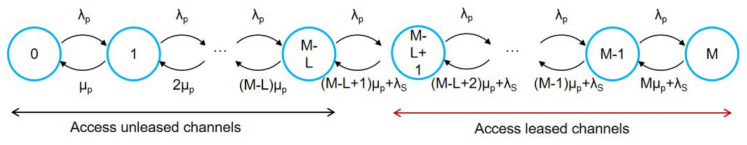
SU network capacity versus λP.

**Figure 8 sensors-20-03800-f008:**
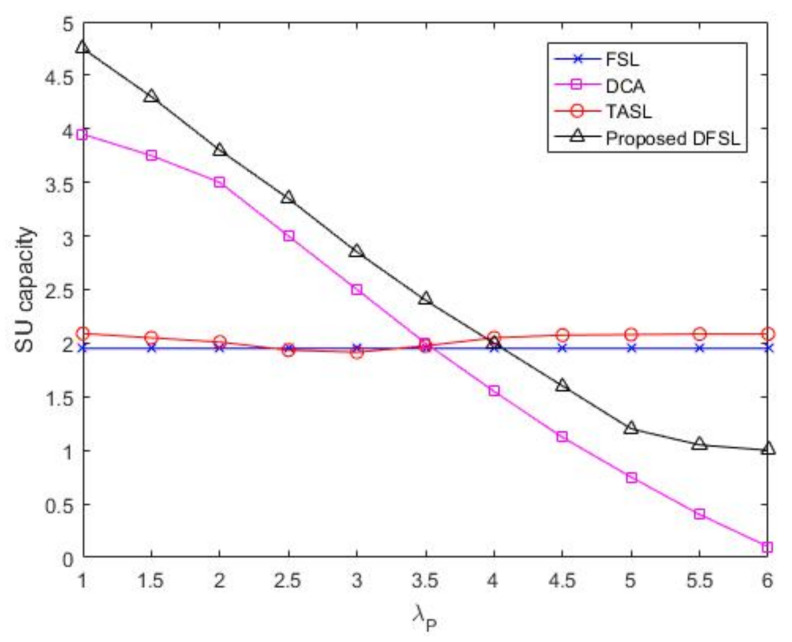
SU network capacity versus $\lambda_{P}$.

**Figure 9 sensors-20-03800-f009:**
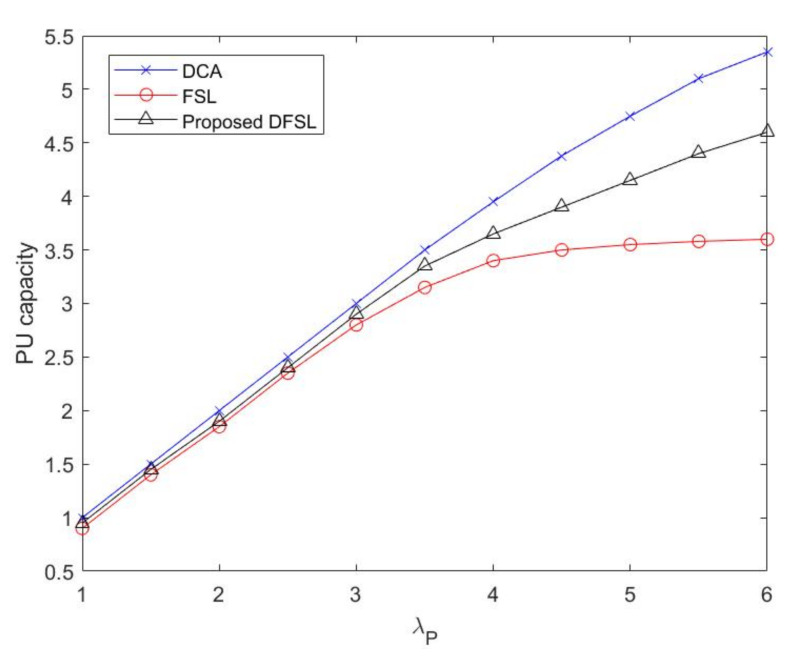
PU network capacity versus $\lambda_{P}$.

**Figure 10 sensors-20-03800-f010:**
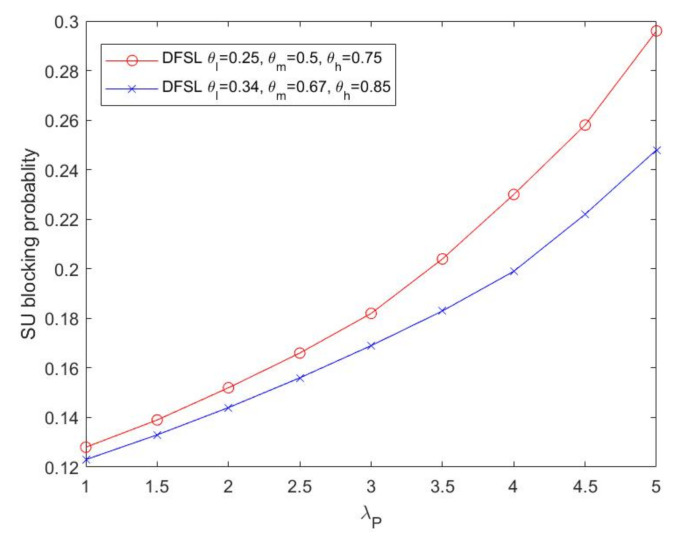
SU blocking probability versus $\lambda_{P}$.

**Figure 11 sensors-20-03800-f011:**
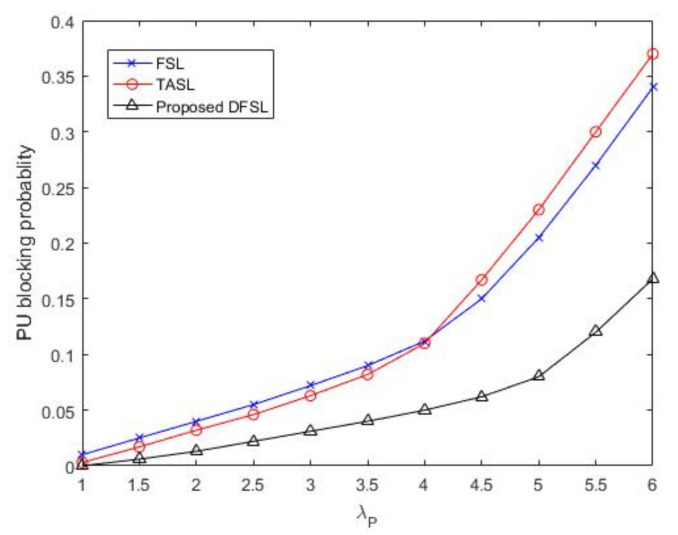
PU blocking probability versus $\lambda_{P}$.

**Figure 12 sensors-20-03800-f012:**
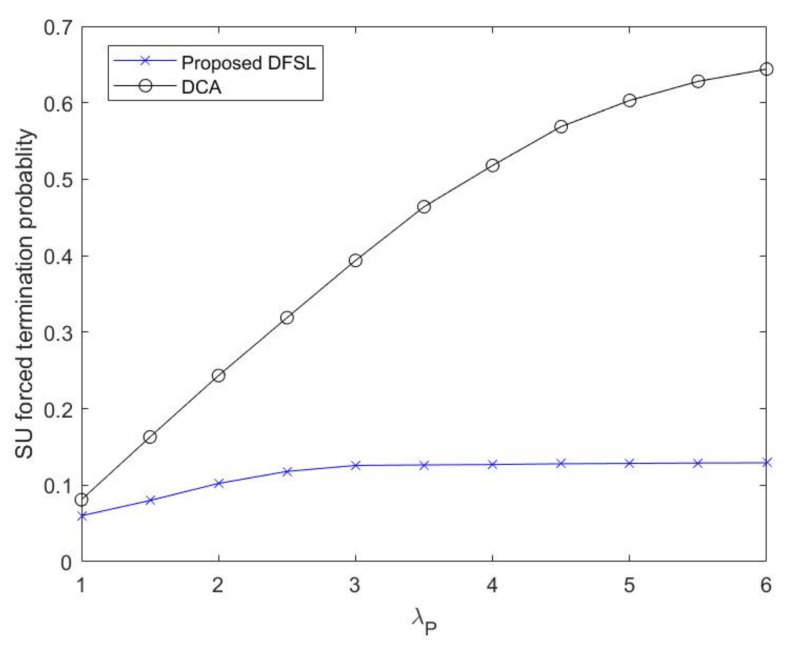
SU forced termination probability with $\lambda_{P}$.

**Table 1 sensors-20-03800-t001:** Transitions of DFSL when a PU arrives.

Event	Destination State	Transition Rate	Conditions
PU arrival. There is a idle channel in N-CRN.	($y_{n}+1,j_{n},y_{l},j_{l}^{1},\ldots ,j_{l}^{i},\ldots ,j_{l}^{V},y_{q},j_{q}$)	$\lambda_{P}$	$\begin{array}{l} b{}_{n}\left(X\right)<M-L,\\ \theta_{l}<\frac{\varepsilon +\eta +\phi }{M}\leq \theta_{m} \end{array}$
PU arrival. There is no idle channel in N-CRN. A SU in N-CRN is forced terminated.	($y_{n}+1,j_{n}-1,y_{l},j_{l}^{1},\ldots ,j_{l}^{i},\ldots ,j_{l}^{V},y_{q},j_{q}$)	$\lambda_{P}$	$\begin{array}{l} b{}_{n}\left(X\right)=M-L,\\ j_{n}>0,\\ b{}_{l}\left(X\right)=L \end{array}$
PU arrival. There is no idle channel and SU in N-CRN. New PU accesses in L-CRN.	($y_{n},j_{n},y_{l}+1,j_{l}^{1},\ldots ,j_{l}^{i},\ldots ,j_{l}^{V},y_{q},j_{q}$)	$\lambda_{P}$	$\begin{array}{l} b{}_{n}\left(X\right)=M-L,\\ j_{n}=0,b{}_{l}\left(X\right)<L \end{array}$
PU arrival. New PU is buffered in queue.	($y_{n},j_{n},y_{l},j_{l}^{1},\ldots ,j_{l}^{i},\ldots ,j_{l}^{V},y_{q}+1,j_{q}$)	$\lambda_{P}$	$\begin{array}{l} b{}_{n}\left(X\right)=M-L,\\ j_{n}=0,b{}_{l}\left(X\right)=L,\\ y_{q}<Q_{PU} \end{array}$
PU arrival. New PU is blocked.	($y_{n},j_{n},y_{l},j_{l}^{1},\ldots ,j_{l}^{i},\ldots ,j_{l}^{V},y_{q},j_{q}$)	$\lambda_{P}$	$\begin{array}{l} b{}_{n}\left(X\right)=M-L,\\ j_{n}=0,b{}_{l}\left(X\right)=L,\\ y_{q}=Q_{PU} \end{array}$

**Table 2 sensors-20-03800-t002:** Transitions of DFSL when a PU departs.

Event	Destination State	Conditions
PU in N-CRN depart.	($y_{n}-1,j_{n},y_{l},j_{l}^{1},\ldots ,j_{l}^{i},\ldots ,j_{l}^{V},y_{q},j_{q}$)	$\begin{array}{l} j_{q}=0\;or\;b{}_{l}\left(X\right)<L;\\ y_{q}=0 \end{array}$
PU in N-CRN depart. A PU in queue accesses.	($y_{n},j_{n},y_{l}-1,j_{l}^{1},\ldots ,j_{l}^{i},\ldots ,j_{l}^{V},y_{q},j_{q}$)	$y_{q}>0$
PU in N-CRN depart. A ongoing PU handover from L-CRN to N-CRN.	($y_{n},j_{n},y_{l}-1,j_{l}^{1},\ldots ,j_{l}^{i},\ldots ,j_{l}^{V},y_{q},j_{q}$)	$\begin{array}{l} j_{q}=0,y{}_{l}>0,\\ y_{q}=0 \end{array}$
PU in N-CRN depart. A SU in queue accesses.	($y_{n}-1,j_{n},y_{l},j_{l}^{1},\ldots ,j_{l}^{i},\ldots ,j_{l}^{V},y_{q},j_{q}-1$)	$\begin{array}{l} j_{q}>0;\;y{}_{l}=0;\\ b{}_{l}\left(X\right)=L;y_{q}=0. \end{array}$
PU in L-CRN depart.	($y_{n},j_{n},y_{l}-1,j_{l}^{1},\ldots ,j_{l}^{i},\ldots ,j_{l}^{V},y_{q},j_{q}-1$)	$\begin{array}{l} j_{q}=0;y_{q}=0 \end{array}$

**Table 3 sensors-20-03800-t003:** Transitions of DFSL when a SU arrives.

Event	Destination State	Transition Rate	Conditions
SU arrival. There is a idle channel in L-CRN.	($y_{n},j_{n},y_{l},j_{l}^{1},\ldots ,j_{l}^{i}+1,\ldots ,j_{l}^{V},y_{q},j_{q}-1$)	$\lambda_{S}$	$b{}_{l}\left(X\right)<L;j_{q}=0$
SU arrival. There is no idle channel in L-CRN. A aggregating SU donates a channel for new SU.	($y_{n},j_{n},y_{l}-1,j_{l}^{1},\ldots ,j_{l}^{i},\ldots ,j_{l}^{V},y_{q},j_{q}$)	$\lambda_{S}$	$\begin{array}{l} b{}_{l}\left(X\right)=L;\\ y_{l}>0;\;j_{q}=0;\\ b{}_{n}\left(X\right)=M-L;\\ m=\max\{i|j_{l}^{i}>0,\\ 1\leq i\leq V\} \end{array}$
SU arrival. There is no available channel in L-CRN. A PU in L-CRN is forced terminated.	($y_{n},j_{n},y_{l}-1,j_{l}^{1},\ldots ,j_{l}^{i},\ldots ,j_{l}^{V},y_{q},j_{q}$)	$\lambda_{S}$	$\begin{array}{l} b{}_{l}\left(X\right)=L;\\ y_{l}>0;\;j_{q}=0;\\ b{}_{n}\left(X\right)=M-L \end{array}$
SU arrival. There is no available channel and PU in L-CRN. New SU accesses in N-CRN.	($y_{n}-1,j_{n},y_{l},j_{l}^{1},\ldots ,j_{l}^{i},\ldots ,j_{l}^{V},y_{q},j_{q}-1$)	$\lambda_{S}$	$\begin{array}{l} b{}_{l}\left(X\right)=L;\\ y_{l}=0;\;j_{q}=0;\\ b{}_{n}\left(X\right)<M-L \end{array}$
SU arrival. New SU is buffered in queue.	($y_{n},j_{n},y_{l}-1,j_{l}^{1},\ldots ,j_{l}^{i},\ldots ,j_{l}^{V},y_{q},j_{q}-1$)	$\lambda_{S}$	$\begin{array}{l} b{}_{l}\left(X\right)=L;\\ y_{l}=0;\;\\ b{}_{n}\left(X\right)=M-L;\\ j_{q}<Q_{SU} \end{array}$
SU arrival. New SU is blocked.	($y_{n},j_{n},y_{l}-1,j_{l}^{1},\ldots ,j_{l}^{i},\ldots ,j_{l}^{V},y_{q},j_{q}-1$)	$\lambda_{S}$	$\begin{array}{l} b{}_{l}\left(X\right)=L;\\ y_{l}=0;\;\\ b{}_{n}\left(X\right)=M-L;\\ j_{q}=Q_{SU} \end{array}$

**Table 4 sensors-20-03800-t004:** Transitions of DFSL when a SU departs.

Event	Destination State	Conditions
SU depart. There is a idle channel in L-CRN.	($y_{n},j_{n},y_{l},j_{l}^{1},\ldots ,j_{l}^{i}-1,\ldots ,j_{l}^{V},y_{q},j_{q}$)	$\begin{array}{l} j_{q}=0;y_{q}=0;\\ j_{n}=0 \end{array}$
SU depart. There is no idle channel in L-CRN. A aggregating SU donates a channel for new SU.	($y_{n},j_{n},y_{l},j_{l}^{1},\ldots ,j_{l}^{i}-1,\ldots ,j_{l}^{V},y_{q},j_{q}-1$)	$j_{q}>0$
SU depart. There is no available channel in L-CRN. A PU in L-CRN is forced terminated.	($y_{n},j_{n}-1,y_{l},j_{l}^{1},\ldots ,j_{l}^{i}-1,\ldots ,j_{l}^{V},y_{q},j_{q}$)	$\begin{array}{l} j_{q}=0;j_{n}>0 \end{array}$
SU depart. There is no available channel and PU in L-CRN. New SU accesses in N-CRN.	$\begin{array}{l} (y_{n},j_{n},y_{l},j_{l}^{1},\ldots ,j_{l}^{i}-1,\ldots ,j_{l}^{r+i}-1,\\ \ldots ,j_{l}^{V},y_{q},j_{q}) \end{array}$	$\begin{array}{l} j_{n}=0;j_{q}=0;\\ r=\min\{i|j_{l}^{i}>0,\\ 1\leq i<V\} \end{array}$
SU depart. New SU is buffered in queue.	($y_{n},j_{n}-1,y_{l},j_{l}^{1},\ldots ,j_{l}^{i},\ldots ,j_{l}^{V},y_{q},j_{q}$)	$\begin{array}{l} j_{q}=0\;or\\ \;b{}_{l}\left(X\right)<L \end{array}$
SU depart. New SU is blocked.	($y_{n},j_{n},y_{l},j_{l}^{1},\ldots ,j_{l}^{i},\ldots ,j_{l}^{V},y_{q},j_{q}-1$)	$\begin{array}{l} j_{q}>0;b{}_{l}\left(X\right)=L \end{array}$

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
