# Peer review of "Dynamic Flow-Adaptive Spectrum Leasing with Channel Aggregation in Cognitive Radio Networks"

_sensors, 2020, doi:10.3390/s20133800_

Round 1

Reviewer 1 Report

In this paper, the authors propose an adaptive leasing algorithm for cognitive radio networks, which adjusts the portion of leased channels based on the number of ongoing and buffered PU flows. The topic is timely, however there are some flaws that the users should take into account.

1) The concept of spectrum leasing is not well motivated. The incentives why a PU would be willing to lease part of its spectrum should be highlighted and justified.

2) This work has some strong assumptions. For instance, the authors assume perfect spectrum sensing, and thus their analysis totally ignores this feauture, which, however, is a major thing for cognitive radio networks.

3) In terms of performance evaluation, the authors only compare their work with their own reference algorithms. However, given that there is a plethora of algorithms in the literature, the authors are encouraged to perform a comparison with sophisticated state-of-the-art solutions.

4) Another very important metric to be studied is energy efficiency. The authors are encouraged to include some discussion on how the proposed solutions would behave in terms of energy efficiency and also include some related works as a reference for future work in the introduction, such as

“Performance Analysis of a Cognitive Radio Contention-Aware Channel Selection Algorithm," IEEE Transactions on Vehicular Technology, vol. 64, no. 5, pp. 1958-1972, May 2015.

"Sensing Time Optimization and Power Control for Energy Efficient Cognitive Small Cell With Imperfect Hybrid Spectrum Sensing," in IEEE Transactions on Wireless Communications, vol. 16, no. 2, pp. 730-743, Feb. 2017

5) The quality of presentation needs to be considerably improved. There are several typos and grammatical errors, such as: "the proposed strategy effectively improve", "while significantly reduce", "have been regarding as promising technology", "the proposed DFSL effectively enhance the network capacity, and improve", "the role as a coordinator", "are presented to evaluates" and many others. Moreover, some acronyms are not defined the first time they appear in the text, such as CTMC.

Author Response

Thank you very much.

Your valuable comments help to improve the quality of the paper. We have thought over all the comments and made our best effort to revise the paper accordingly.

Enclosed please find the one-to-one responses to the reviewer’ comments. Please kindly let us know if there are any additional comments/concerns.

Reviewer 2 Report

The topic is not novel and a lot of works have investigated similar topics. I have the following major comments.

  1. What is “flow”? Data flow or something else? To the reviewer, flow is generally utilized in mesh networks connecting with multiple routers. However, the system model does not present a mesh network.
  2. The continuous time Markov chain is developed to model DFSL. What is the formal relationship of the transition rate and the dynamic characteristics of PU and SU (e.g., arrival rate and numbers)?
  3. What is the algorithmic complexity of DFSL? Will the increase of the number of users and channels affect the calculation efficiency?
  4. The simulation setup is not clearly described. For example, what is the channel model? Is there interference from other system? Is this a system-level simulation?
  5. DCA and FSL are not depicted or even given a citation. So how to follow the algorithm.
  6. The references are not new. Please introduce the most recent related works.
  7. There are still grammar errors. The authors should check out the whole paper.

Author Response

(The authors gave the same response as above.)

Round 2

Reviewer 1 Report

The authors took into account all my comments, and thus, I propose the publication of this manuscript in its current form.

Reviewer 2 Report

I have no further comments.